# Integrated Bioinformatic Investigation of EXOSCs in Hepatocellular Carcinoma Followed by the Preliminary Validation of EXOSC5 in Cell Proliferation

**DOI:** 10.3390/ijms232012161

**Published:** 2022-10-12

**Authors:** Yujing Zhang, Xinyue Yang, Yang Hu, Xin Huang

**Affiliations:** Key Laboratory of Molecular Epidemiology of Hunan Province, School of Medicine, Hunan Normal University, Changsha 410081, China

**Keywords:** EXOSCs, HCC, bioinformatics analysis, EXOSC5, STAT3

## Abstract

The Exosome complex (EXOSC) is a multiprotein complex that was originally discovered as the machinery of RNA degradation. Interestingly, recent studies have reported that EXOSC family members (EXOSCs) are associated with various human diseases, including cancers. It will be interesting to investigate whether EXOSCs are related to the processes of hepatocellular carcinoma (HCC). In this study, multiple public databases and experimental validation were utilized to systemically investigate the role of EXOSCs, especially EXOSC5, in HCC. It is worth considering that the mRNA and protein levels of many EXOSCs were elevated in HCC, although there were some differences in the results from different database analyses. The over-expression of EXOSCs could predict HCC to some extent, as evidenced by the positive correlation between the elevated EXOSCs and alpha fetoprotein (AFP) levels, as well as with a high accuracy, as shown by the receiver operating characteristic curve analysis. Additionally, higher mRNA expressions of specific EXOSCs were significantly related to clinical cancer stage, shorter overall survival and disease-free survival in HCC patients. A moderate mutation rate of EXOSCs was also observed in HCC. Furthermore, a gene functional enrichment analysis indicated that EXOSCs were mainly involved in the metabolism of RNA. Moreover, we revealed that the expression of EXOSCs is remarkably related to immune cell infiltration. Finally, EXOSC5 was upregulated in HCC tissues and cell lines, promoting cell growth and proliferation via activated signal transducer and activator of transcription 3 (STAT3). The bioinformatic analyses, following verification in situ and in vitro, provided a direction for further functions and underlying mechanism of EXOSCs in HCC.

## 1. Introduction

Hepatocellular carcinoma (HCC) is one common malignant tumor of the digestive system, and is a major threat to public health, especially in China [1]. It is a heterogeneous malignant tumor and has the sixth highest morbidity and fourth highest mortality in the world [2]. Currently, surgical resection and liver transplantation may be the best ways to treat HCC, but the 5-year overall survival rate is only 7% [3]. Although great advances have been made in the experimental and clinical treatment of HCC, its diagnosis is very difficult, and its prognosis is poor due to complicated etiology and multiple genes’ alteration in individuals [4,5,6]. Therefore, a better understanding of the molecular mechanisms and highly reliable biomarkers of HCC is of great significance to promoting the diagnosis and development of therapeutic strategies for HCC.

RNA levels are regulated during cell differentiation and development in responses to intrinsic and extrinsic stimulation. Recently, RNA degradation has begun to be regarded as a significant mechanism of RNA regulation in the control of the gene expression involved in tumor progression [7]. Ample information accumulated on RNA degradation has been studied intensely [8]. It is noteworthy that the exosome complex (EXOSC) plays a central role in the degradation process of multiple *RNAs* both in the nucleus and cytoplasm [9]. In humans, the EXOSC is composed of ten evolutionarily conserved subunits, including the barrel-shaped core (EXOSC4-9 subunits) and three-subunit cap (EXOSC1-3) covering its surface, while EXOSC10 is bound to the catalytically inactive core and cap subunits of the EXOSC [10]. Apart from the RNA degradation function of the EXOSC, its components are also associated with carcinogenesis. It has been reported that EXOSC1 could cleave single-stranded DNA and sensitize human kidney renal clear cell carcinoma cells to poly (ADP-ribose) polymerase inhibitors [11]. Additionally, the enhancing expression of EXOSC2 directly regulated by tRNAGluUUC is closely related to breast cancer metastasis [12], while EXOSC3 and EXOSC4 can trigger the development of colonic and colorectal cancer, respectively [13,14]. Similarly, EXOSC5, EXOSC8, EXOSC9 and EXOSC10 are also remarkably involved in the progression of various cancers, but there is no evidence to show the association between EXOSC6/7 and cancers at present. Interestingly, growing evidence suggests the role of EXOSC5 in the progression of cancers [15,16]. It has been reported that EXOSC5 is the no-catalytic component of EXOSC, while it can directly interact with the zinc-finger antiviral protein to degrade the target RNA. Currently, most studies have reported that up-regulation of EXOSC5 in various epithelial and hematopoietic tumor and cell lines [15]. Recently, Pan et al. have indicated that the upregulation of EXOSC5 increases cell proliferation in colorectal cancer via the ERK and AKT signaling pathways [15], suggesting the effects of EXOSC5 on digestive cancers. However, whether EXOSC5 is implicated in the process of HCC is still unclear. Thus, we aim to indicate the role and underlying mechanism of EXOSC5 in HCC via public databases and preliminarily validation from experiments in situ and in vitro.

A previous study demonstrated that the signal transducer and activator of transcription 3 (STAT3) signaling pathway played a key role in the proliferation of gastric cancer mediated by EXOSC5 [10]. STAT3 is one of the members of the STAT family, which was first identified from a mouse liver cDNA library in the study of IL-6 signaling (NF-κB and STAT3) [17,18]. Several studies have found that STAT3 may be critical for the development of chemically induced HCC because it was a transcriptional factor closely related to immune responses, inflammation and tumorigenesis [19]. However, the events that regulate STAT3 activation in human HCC are not known.

In this study, based on a variety of public databases and experimental analysis, for the first time we extended the research field of HCC with the purpose of determining the values of EXOSC family members (EXOSCs) in HCC. Moreover, the role of EXOSC5 in the cell growth of HCC and the underlying mechanism were also determined. The findings in this report shed light on the relationship between EXOSCs, especially EXOSC5, and HCC. It also provided a potential target for the diagnosis of HCC and indicated the possible mechanism of EXOSC5 in promoting HCC proliferation.

## 2. Results

### 2.1. Differential Expression of EXOSCs in HCC Patients

To investigate the distinct expression of EXOSCs in HCC, the TCGA database was used to analyze the mRNA levels of EXOSCs in tumor and normal tissues. The result in Figure 1A showed that the transcriptional levels of EXOSCs besides EXOSC6 were significantly elevated in HCC tissues compared with normal tissues. Interestingly, the results from the GEPIA dataset showed that only the EXOSC4/5 transcriptional levels were different between HCC and normal tissues (Figure 1B), which was different from what was shown in the TCGA analysis. Furthermore, the expression of EXOSCs between tumors and normal tissues in various cancers was evaluated by utilizing the TIMER database, showing the differential expression of EXOSCs in many cancers (Appendix A). Following the analysis of EXOSCs mRNA levels, the protein expression of these members was also assessed using clinical proteomic tumor analysis consortium (CPTAC) and the Human Protein Atlas database. As shown in Figure 2, the protein expression of EXOSCs was respectively higher in HCC. However, the immunohistochemical information of EXOSC6 in HCC and normal tissues was not found in the database. In summary, EXOSCs were remarkably expressed in HCC with certain differences in various databases, which need to be further analyzed.

### 2.2. The Diagnostic Value of EXOSCs in HCC

Given that the EXOSCs were remarkably elevated in HCC, and that their up-regulation was related to the clinical characteristics, the correlation between EXOSCs and AFP was analyzed due to the fact that it is a valuable diagnostic marker for HCC. According to the instruction of Guidelines for Diagnosis and Treatment of Primary Liver Cancer (2022), AFP is a common biomarker for the diagnosis of liver cancer and curative effect. Serum AFP ≥ 400 ng/mL is highly regarded as liver cancer, excluding pregnancy, chronic or active liver disease, germ gland embryogenic tumor and digestive tract tumor, we then analyzed the correlation between EXOSCs and AFP (400 ng/mL). Figure 3A shows that EXOSCs were positively correlated with serum AFP. Moreover, receiver operating characteristic curve (ROC) analysis showed that the average the area under the ROC curve (AUC) of EXOSCs was 0.887, which was calculated for the evaluation of the prediction accuracy of EXOSCs in HCC, indicating relatively high specificity and sensitivity of the prognosis (AUC > 0.80) in HCC except EXOSC6 (AUC = 0.635) (Figure 3B). The correlation between the expression of differentially expressed EXOSCs and the pathological stage of HCC patients was analyzed. The EXOSC2/3/5/6/7/8/9/10 were highly related to Stage II and III HCC, while EXOSC1/4 were not markedly different (Figure 3C).

### 2.3. Prognostic Potential of EXOSCs in HCC

We then investigated whether the differentially expressed EXOSCs were associated with HCC prognosis. The GEPIA and TCGA databases, as well as the Kaplan–Meier plotter, were used to analyze the correlation of EXOSCs with the clinical outcomes of HCC. As shown in Figure 4A, using GEPIA database, the overall survival (OS) curve showed that EXOSCs with a high mRNA expression had an obvious effect on the short OS of HCC, except EXOSC6/8. Furthermore, the analysis from the TCGA database Kaplan–Meier plotter also indicated the prognostic values of EXOSCs in patients with HCC, showing a high expression of EXOSC2/3/4/5/6/7/9/10 was significantly correlated with short OS in HCC (Appendix AA). In addition, results from the Kaplan–Meier plotter showed a close relationship between the up-regulated expression of EXOSC1/2/3/5/6/7/9/10 and worse prognosis of HCC patients (Appendix AB). Meanwhile, GEPIA database also showed that HCC patients with high transcriptional levels of EXOSC1/2/7/8/9/10 were markedly associated with short disease-free survival (DFS) (Figure 4B). These data suggested that mRNA expressions of EXOSCs were remarkedly related to the prognosis of HCC patients and some of them may be a biomarker with which to estimate the survival of HCC patients.

### 2.4. The Genetic Alterations and Correlation of EXOSCs in HCC

The genetic alterations of EXOSCs were assessed by the cBioPortal database based on mutation and copy number alteration data from different HCC studies. As shown in Figure 5A, a moderate mutation rate of EXOSCs was observed in HCC patients. There are three types of genomic alterations, including amplification, deep deletion and structural variant. EXOSC4 ranked as the gene with the highest genetic alterations, with a mutation rate of 10%. The neighbor gene of EXOSC4 with the most frequent alterations was EXOSC10 (2.5%), while other members had lower mutation rates (less than 1%) (Figure 5B). However, Kaplan–Meier plotter and log-rank test analysis showed that EXOSC4/10 mutations had no significant correlation with OS and DFS in HCC (Figure 5C–F). Furthermore, the mRNA expressions of EXOSCs were analyzed to indicate their correlations with each other via the cBioPortal. The result in Figure 5G showed that here was, for most part, an obviously positive correlation between any two EXOSCs, while there was a negative correlation between EXOSC4 and EXOSC10.

### 2.5. Predict Functions and Pathways of EXOSCs Genes

Subsequently, a functional analysis of these interacting genes was examined via Metascape and the TCGA database. Gene oncology (GO) and Kyoto Encyclopedia of Genes and Genomes (KEGG) analysis showed three signal pathways (BP), three cell components (CC), three biological processes (MF) and one KEGG, respectively (Figure 6A). We found that EXOSCs were involved in exoribonuclease activity, producing 5′-phosphomonoesters, 3′-5′-exoribonuclease activity, exoribonuclease complex, etc., via GO analysis. The KEGG analysis showed that EXOSCs were mainly involved in RNA degradation. The top 20 clusters with the representative enriched items were showed in Figure 6B. EXOSCs and their related genes were mainly enriched in the following aspects: metabolism of RNA, processing of capped intron-containing pre-mRNA, complement and PPAR signaling pathway and so on. In addition, Metascape was utilized to assess the network. The results in Figure 6C present enriched term as a colored node.

### 2.6. Immune Cell Infiltration of EXOSCs in HCC

In the present study, the TIMER database was used to analyze the association between the differentially expressed EXOSCs and immune cell infiltration in HCC patients. Figure 7 indicated that EXOSCs, except EXOSC4, were significantly correlated with diverse immune cells, including B cells, CD4+ T cells, CD8+ T cells, neutrophils, macrophages and dendritic cells in HCC patients. Furthermore, single sample gene set enrichment analysis (ssGSEA) with Spearman’s rank correlation was utilized to assess the correlation between EXOSCs and 24 types of immune cell infiltration. The results in Appendix A showed that differentially expressed EXOSCs were all positively correlated with Th12 cells, while being negatively correlated with dendritic cells, Neutrophils and Th17 cells. Next, we analyzed the correlation of between EXOSCs and immunosuppressive molecules, MHC molecules, chemokines and chemokine receptors in HCC using the TISIDB. The results showed that upregulated expression of EXOSCs was associated with an increase in the expression of immunosuppressive molecules such as CTLA4, MHC molecules such as HLA-A, chemokine receptors such as CCR3 and chemokines such as CCL3 (Figure 8A–D), providing important information for predicting a potential therapeutic target for HCC. Altogether, these results confirmed the fact that EXOSCs played a pivotal role in the immune regulation of HCC patients.

### 2.7. The Expression of EXOSC5 in HCC Tumor Tissues and Cell Lines

Based on the above bioinformatics analysis, we further mainly investigated the role of EXOSC5 in HCC. The mRNA expression of EXOSC5 was determined by qRT-PCR in 15 matched HCC and adjacent non-tumor tissues. As shown in Figure 9A, the EXOSC5 mRNA level was significantly increased. Furthermore, the elevated EXOSC5 expression in HCC tissues was validated by immunohistochemistry (IHC) and western blot (Figure 9B,C). In addition, to further confirm the differentially expressed levels of EXOSC5 in HCC, the protein and mRNA levels of EXOSC5 in L02 hepatocytes and HCC cell lines (SMMC7721, Huh7, LM3, HepG2 cells) were detected by western blot and qRT-PCR. As shown in Figure 8D, E, the results indicated that EXOSC5 was significantly increased in HCC cell lines, especially in SMMC772 and Huh7 cells. These data were consistent with the bioinformatic analysis, which suggested the elevated expression of EXOSC5 in HCC.

### 2.8. EXOSC5 Promotes Cell Proliferation via Enhancing the Phosphorylation of STAT3

EXOSC5 siRNA was used to inhibit the expression of EXOSC5 in SMMC7721 cells to investigate its function and underlying mechanisms in HCC. The result from qRT-PCR in Figure 10A showed that siRNA-1 had the best suppression efficiency of EXOSC5. Similarly, the protein levels of EXOSC5 also validated the inhibitory effects of siRNA-1 (Figure 10B). Subsequently, siRNA-1 was chosen to inhibit EXOSC5 in SMMC7721 cells. MTT assay and colony formation indicated the role of EXOSC5 in the proliferation of SMMC7721 cells, evidenced by the downregulated absorbance (OD) value and number of cell cloning (Figure 10C,D). Meanwhile, the decreased expression of Ki67, which was a biomarker of cell growth, was also found (Figure 10E). To find the regulatory mechanism of EXOSC5 in the cell growth of SMMC7721 cells, the phosphorylation and total protein levels of STAT3 were detected by western blot. As shown in Figure 10F, EXOSC5 knockdown inhibited the expression and phosphorylation of STAT3.

To further confirm the role of activated STAT3 in HCC, we assayed the protein levels of STAT3 in HCC and adjacent tissues. Figure 11A,B showed the upregulated expression of STAT3 in HCC tissue and cell lines. We then chose the specific inhibitor, S31-201, to treat SMMC7721 cells. As shown in Figure 11C,D, S31-201 did not influence the mRNA levels of STAT3, but it could inhibit its phosphorylation. Results also showed that the inhibition of STAT3 phosphorylation could further aggravate the decreased cell growth in EXOSC5 knockdown cells (Figure 11E,F). The above results suggested that EXOSC5 plays a role in partially promoting the proliferation of HCC by activating STAT3.

## 3. Discussion

HCC is a serious problem with an increased incidence in both developing and developed countries. Many factors, including virus infection and cirrhosis, have been proven to be closely related to HCC development [20]. HCC is typically characterized by a low rate of early diagnosis and poor prognosis. Currently, elevated levels are regarded as a conventional biomarker for the diagnosis of HCC in clinical practice [21], but it has its own limitations due to the fact that many advanced HCC patients are AFP-negative. Fortunately, systemic treatment options are available for advanced HCC in recent years with the advent of new therapeutic targets and immunotherapy, bringing new hope to HCC patients [22]. Although advances have been made in the therapeutic management of HCC, the prognosis for it remains unknown and without established biomarkers. Therefore, HCC-related genes or biomarkers have yet to be identified to clarify the underlying mechanisms of its diagnosis, prognosis and treatment. In the present study, we comprehensively analyzed, for the first time, EXOSCs in terms of expression, mutation, prognostic value and immune cell infiltration in HCC. Even with evidence that EXOSCs have been confirmed to be associated with the development of many digestive cancers, distinct roles of EXOSCs in HCC remained to be investigated. Furthermore, the function of EXOSC5 and related mechanisms were verified by the cell assay and immunohistochemical staining.

Being important components in RNA degradation, EXOSCs are crucial for controlling coding and non-coding RNA levels, profoundly participating in the elimination of defective RNAs in all organisms [23]. In addition to DNA damage, results from the past couple of decades have unequivocally elucidated that RNA degradation takes part in the development and progression of multiple cancers [24]. Therefore, EXOSCs that serve as an indispensable part of RNA stability may be associated with the process of cancers, making them an attractive target for cancer diagnosis and treatment. For instance, compared with healthy cells, cancer cells (i.e., MDA-MB-231, MCF-7 and Hela cells) showed higher activity of EXOSC9 [25]. Additionally, functional experiments also confirmed the oncogenic roles of EXOSC8 in colorectal carcinoma [26]. Results from our study showed that the mRNA expressions of EXOSCs, except EXOSC6, were differentially expressed in HCC, especially EXOSC4 and EXOSC5. Not only were the over-expression of EXOSCs proteins found, but a marked correlation between up-regulated EXOSC2/3/6/7/8/9/10 and clinical tumor stage was demonstrated. Moreover, we analyzed the expression of EXOSCs in multiple tumors relative to normal tissues. The results pointed out that EXOSCs were remarkably upregulated in various tumors, including cholangiocarcinoma (CHOL), kidney chromophobe (KICH) cancer, kidney renal clear cell carcinoma (KIRC), HCC, etc. (Appendix A). These data suggested the higher levels of EXOSCs in tumors, which were similar to those stated in previous reports.

Besides this, the correlation between EXOSCs and AFP, as well as the ROC, were used to analyze the ability of differentially expressed EXOSCs to predict HCC, suggesting that EXOSCs had a certain accuracy in predicting the outcome of HCC. Similarly, previous studies also identified the roles of EXOSC4 and EXOSC5 in predicting the prognosis of colorectal cancer and renal cell carcinoma, respectively [14,16]. At the same time, we also found that the high expression of EXOSCs was correlated with shorter OS and DFS, although there were some differences in the results from different databases analysis, indicating that differentially expressed EXOSCs influenced the prognosis of HCC patients. As a result, we speculated that EXOSCs may serve as potential diagnostic and prognostic markers for HCC. However, further verifications are required in order to discuss which EXOSC family members are directly related to poor prognosis of HCC.

Gene mutation is essential for biological evolution and tumorigenesis. There are currently few studies on the gene mutations of EXOSCs in the development and progression of cancer, while evidence has indicated that EXOSCs were closely related to many gene mutations [11]. A previous review reported that the mutation of EXOSC2/3/8 could underly dysfunction and induced various diseases [27]. Meanwhile, our genetic analysis indicated moderate genetic alterations of EXOSCs in HCC, which was consistent with previous reports to some extent. However, the association between EXOSCs mutations and HCC needs to be further explored. Array CGH and PPI mapping integrating study has proved that EXOSC1/3/8 were predicted to be the functional interaction factors in the network of the 6q23.3 locus in melanoma [28]. Our data also uncovered the correlation between EXOSCs family members, and the similar results with previous study further prompted the significance of the interaction of EXOSCs family members. In this study, we also showed that EXOSCs participated in various biological processes, especially the metabolism of RNA, playing a key role in the development of multiple cancers. This not only confirmed the role of EXOSCs in cancer development [29], but also provided clues for the future research on the function of EXOSCs in HCC.

The tumor microenvironment (TME) is a key factor in tumor progression and recurrence, causing it to become an increasingly popular topic. Growing evidence indicates that innate and adaptive immune cells could release inflammatory mediators such as cytokines, chemokines, growth factors and proteolytic enzymes, as well as activate transcription factors (NF-κB, STAT3, etc.), contributing to tumor progression when present in the TME [30]. Therefore, immune cells are of significance in both clinical outcomes and the response to immunotherapy [31,32]. It has been reported that adaptive immune cells, such as CD8+ T cells, Th17 cells and B cells could stimulate HCC development, while adaptive immunity promotes immune surveillance to eradicate early HCC [33]. According to these findings, we found that the expression of EXOSCs might be correlated with the infiltration of B cells, CD4+ T cells, CD8+ T cells, neutrophils, macrophages and dendritic cells, indicating that EXOSCs may reflect the immune status alongside disease prognosis. Thus, the study provides new insights into understanding the role of EXOSCs in the immune cell infiltration of HCC and provides detailed information to design new immunotherapies.

On the basis of these results from public databases, which suggested the significantly different expression and diagnostic, as well as prognostic, roles of EXOSC5, which are similar conclusions to those found in a previous report, exhibiting the potential clinical predictive and prognostic value of EXOSC5 [15]. EXOSC5, also known as Rrp46p or CML28, is involved in a multitude of cellular RNA processing and degradation events via its exoribonuclease activity, showing an effect on HBV RNA degradation [34,35]. Thus, we decided to preliminarily validate the role of EXOSC5 in HCC proliferation. Yang et al. first identified EXOSC5, which is overexpressed in various tumor cell lines, but not in normal tissues [36]. Wu et al. revealed that the elevation of EXOSC5 was observed in leukemic blasts from patients with acute myelogenous leukemia and chronic myelogenous leukemia blast crisis, while it is barely detected in normal bone marrow and peripheral blood [37]. Additionally, the overexpression of EXOSC5 was also detected in various histological tumors [38]. Although results have highlighted the oncogenic role of EXOSC5 in colorectal cancer, there have been few studies on the role and related molecular mechanism of EXOSC5 in solid tumors. Herein, we found the upregulation of EXOSC5 mRNA and protein levels in HCC tumor tissues, and similar results were also found in HCC cell lines. Having a larger dataset about the expression of other members of EXOSCs family in HCC would offer further insights and more research is necessary in the future regarding EXOSCs. Evidence in our study also indicated that EXOSC5 knockdown suppressed the cell growth and proliferation of SMMC7721 cell lines. As far as we all know, this is the first study to report the role of EXOSC5 in HCC by in situ and in vitro experiments. A previous study reported that the STAT3 signaling pathway was involved in the EXOSC5-regulated proliferation of gastric cancer [39], and the elevated phosphorylation of STAT3 was also shown in HCC tissues and cell lines, which could be inhibited by the down-regulation of EXOSC5 in the present study. Meanwhile, the increased phosphorylation of STAT3 was implicated in cell proliferation that is consistent with the function of STAT3 in cancers [19]. Therefore, we suggested that EXOSC5 played important roles in HCC proliferation through activating STAT3 signaling. However, further studies are warranted to clarify the direct target and underlying mechanism of EXOSC5 in HCC patients.

Taken together, we systematically analyzed the differential expression and prognostic value of EXOSCs in HCC in silico for the first time. Furthermore, we deeply indicated that the over-expression of EXOSC5 was significantly related to proliferation of HCC via the STAT3 signaling pathway. Our study provided preliminary evidence for the role and a possible signaling for EXOSC5 in promoting HCC proliferation, but further research about molecular mechanisms of EXOSC5 in HCC is required.

Nevertheless, some limitations in our study should be recognized. Such as the location of these data only from public databases and lack of larger HCC cohorts. Moreover, other members of EXOSCs were also differentially expressed, except EXOSC5, but we did not provide the expression of these EXOSCs in the current study. In addition, the conclusion of our research only preliminarily confirmed the correlation between higher EXOSCs and HCC, as well as a related signaling of EXOSC5 promoting cell proliferation in vitro. More prospective data and larger experiments using tumor tissues, cells and animal models are required to confirm clinical suitability of EXOSCs in HCC.

## 4. Materials and Method

### 4.1. Data Resources for Clinical and Pathological Information of EXOSCs in HCC

In the study, we analyzed the expression of EXOSCs in HCC through multiple databases. TCGA (https://www.cancer.gov/) (accessed on 22 July 2022) is a landmark cancer genomics program that provides molecular characterization of tumor samples and matched normal samples of more than 20,000 primary cancers across 33 cancer types. Clinical information on patients with HCC and high-throughput RNA-sequencing data were downloaded from the TCGA database. The transcript expression levels were estimated using the fragment per kilobase per million fragments mapped (FPKM) method in HTSeq. R (version 3.6.3) was used to analyze the expression levels of EXOSCs in tumor and adjacent tissues in the TCGA database and visualized with ggplot2 (3.3.3). Furthermore, other newly developed analytical tools, GEPIA2 (http://gepia2.cancer-pku.cn)(accessed on 22 July 2022) [40] and TIMER2 (http://timer.cistrome.org/) (accessed on 22 July 2022) [41], were applied to assess the mRNA expression of EXOSCs in HCC and normal tissues. GEPIA 2 is a newly developed tool using a standard processing pipeline to provide com-prehensive analysis, such as differential expression analysis, profiling plotting, correlation analysis and patient survival analysis, etc. TIMER2 is commonly used to analyze the relationship between immune infiltration and gene expression, clinical outcome and somatic mutations. Herein, we used it to investigate the differential expression of EXOSCs in multiple tumors. Besides, the total expression of EXOSCs in the Clinical Proteomic Tumor Analysis Consortium (CPTAC) database (https://proteomics.cancer.gov/) (accessed on 22 July 2022) by The University of ALabama at Birmingham CANcer data analysis Portal (Ualcan) (http://ualcan.path.uab.edu) (accessed on 22 July 2022) database, which could quantify and identify the constituent proteins of each tumor sample tumor biospecimen using mass spectrometry [42,43]. We analyzed the data of 379 HCC patients and 50 healthy people in TCGA and 110 healthy people in The Genotype-Tissue Expression (GTEx) (www.gtexportal.org) (accessed on 22 July 2022). The Human Protein Atlas database (HPA) (https://www.proteinatlas.org) (accessed on 22 July 2022) [44,45] is a Swedish initiative launched in 2003 to map all human proteins in cells, tissues and organs using the integration of various omics technologies, including antibody-based imaging, mass spectrometry-based proteomics and transcriptome science. The initiative also provides free access to immunohistochemical images of human-related tumor tissues and corresponding normal tissues. In this study, HPA was used to compare the protein levels of EXOSCs in HCC tumor and normal tissues.

### 4.2. Diagnostic Performance of EXOSCs

The serum AFP levels of HCC patients and normal were obtained by using the TCGA database, which were divided into two groups with a boundary of 400 ng/mL. R (version 3.6.3) was used to analyze the relationship between the expressions of EXOSCs and serum AFP. ggplot2 (3.3.3) was used for visual analysis. Additionally, we used GEPIA2.0 to draw ROC curve and calculated the AUC.

### 4.3. Survival Analysis for EXOSC Family Genes in HCC

The GEPIA2 website was used to analyze the survival data related to different cancer patients in the GTEx database (www.gtexportal.org) (accessed on 22 July 2022). The influence of EXOSCs levels on the prognosis of HCC was analyzed. In addition, according to the median mRNA expression of EXOSCs, all HCC patients were divided into EXOSCs mRNA high and low expression groups from the TCGA database. Finally, the Kaplan–Meier survival curve was drawn by the survminer (version 4.9) package and the survival package (version 3.2-10) to analyze the effects of EXOSCs on the clinical prognosis of HCC patients. Furthermore, the results were verified again through the Kaplan–Meier plotter website (www.kmplot.com/) (accessed on 22 July 2022). The website data comes from Gene Expression Omnibus (GEO), European Genome-phenome Archive (EGA) and TCGA. The Kaplan–Meier survival curve showed the relationship between EXOSCs and OS in HCC patients.

### 4.4. Genomic Mutation and Correlation Analysis of EXOSCs

cBioPortal (http://cbioportal.org) (accessed on 22 July 2022) is an open-access resource that provides data from more than 5000 tumor samples from 20 cancer studies and explores the interaction of multidimensional cancer genomics data sets (The cBioportal: an open platform for exploring multidimensional cancer genomics data). This database was utilized to analyze alterations of EXOSCs in TCGA HCC samples, which further evaluated the correlation of these mutations with important clinicopathological factors.

### 4.5. Correlation and Interaction Analysis of EXOSCs

The TIMER (http://timer.cistrome.org/) (accessed on 22 July 2022) database was applied to analyze the correlation of the expression of EXOSCs in HCC, and the statistical significance was evaluated by a Spearman’s test. In addition, STRING (https://string-preview.org/) (accessed on 22 July 2022) provides users with protein interactions network functional enrichment analysis. Both of these were utilized to investigate the EXOSCs gene–gene and protein–protein interactions.

### 4.6. Functional Enrichment Analysis of EXOSCs

Metascape (http://metascape.org) (accessed on 22 July 2022) is widely used in GO enrichment analysis of BP, CC, MF, as well as in pathway enrichment analysis (Metascape provides a biologist-oriented resource for the analysis of systems-level datasets). Herein, Metascape was used to perform pathway and process enrichment analysis of EXOSCs with the surrounding genes that were remarkably related to the expression of EXOSCs. A subset of enriched terms that were selected was displayed as a network plot, such that we could better understand the relationships between these terms. To further identify the closely related network components, the molecular complex detection (MCODE) algorithm was administered.

### 4.7. Immune Infiltration Analysis of EXOSCs

TIMER (https://cistrome.shinyapps.io/timer/) (accessed on 22 July 2022) is a web-based application that could conduct a comprehensive analysis of the immune cell infiltration. In the present study, this database was used to analyze the correlation between the genes of EXOSCs and immune cells infiltration in HCC. Additionally, the TISIDB (http://cis.hku.hk/TISIDB/) (accessed on 22 July 2022) database was administered to assess the expression of EXOSCs expression in different immune subtypes of HCC patients.

### 4.8. IHC

A total of 15 HCC and adjacent non-tumor tissues were collected from The First Affiliated Hospital of Hunan Normal University, Changsha, China. The tissues were all immediately frozen in liquid nitrogen and stored at −80 °C. The tissues were cut into 5 μm sections before 30 min at RT. The sections were then fixed with 4% PFA for 15 min at RT and washed with PBS. Then, 3% hydrogen peroxide (H_2_O_2_) was used to block the endogenous peroxidase activity for 30 min at RT, and the sections were pre-treated with sodium citrate buffer (pH = 6) for 30 min. The specific primary antibody of EXOSC5 (Proteintech group, Wuhan, China) was used to incubate the sections (1:200) at 4 °C overnight. 3,3-diaminobenzidine tetrahydrochloride (DAB) solution was used to stain the sections following with incubation by HRP-conjugated secondary antibody at 37 °C for 45 min. The sections were counterstained with hematoxylin and observed under a microscope.

### 4.9. Cell Culture

The normal hepatocytes (L02) were a gift from Nanjing Medical University and the HCC cell lines (HepG2, SMMC7721, Huh7, LM3) were purchased from the American Type Culture Collection (Manassas, VA, USA). L02 hepatocytes were cultured by RPMI1640 medium with 10% fetal bovine serum (FBS; Biological Industries, BeitHaemek, Israel) and 100 units/mL penicillin and streptomycin (Invitrogen, Carlsbad, CA, USA). HepG2, SMMC7721, Huh7 and LM3 cells were all cultured in Dulbecco’s modified Eagle’s medium (DMEM) with 10% FBS and 100 units/mL penicillin and streptomycin. The STAT3 inhibitor S31-201 dissolved in DMSO (Selleck Chemicals, Houston, TX, USA) was utilized to suppress STAT3 signaling. These cells were kept at 37 °C in a humidified % CO_2_ incubator.

### 4.10. Transfection of EXOSC5 siRNA in SMMC772 Cells

The SMMC7721 cells were seeded into 6-well cell culture tissue for 6~8 h. Then, 100 nm EXOSC5 siRNA and a negative control (NC) were transferred into SMMC7721 cells using Lipofectamine 2000 (Life Technologies, Carlsbad, CA, USA) according to the manufacturer’s protocol. The sequence of EXOSC5 siRNAs was synthesized by RIBIOBIO (Guangzhou, China). These sequences are as follow: siRNA-1, 5′-GCCAAAATCCGTG CTGAAAATGG-3′, siRNA-2,5′-CAGCAAAGAGATTTTCAACAAGG-3′, siRNA-3, 5′-GGAATGGATGAATCAATAAATT-3′. The knockdown expression of EXOSC5 was determined by qRT-PCR and western blot.

### 4.11. MTT Assay

SMMC7721 cells transfected with EXOSC5 siRNA or treated with S31-201were seeded into 96-well cell culture plates at density of 2 × 10^3^. Then, 20 μL MTT (5 mg/mL) were added into the well of 0 h after 2 h. Subsequently, DMSO was added into each well every 24 h. After 120 h of culture, the absorbance was measured by microplate reader at a wavelength of 490 nm.

### 4.12. Colony Formation Assay

SMMC7721 cells transfected with EXOSC5 siRNA, or treated with S31-201, were seeded into 6-well culture plates at a density of 1 × 10^3^ cells/well. The cells were cultured in an incubator with 5% CO_2_ at 37 °C for 2 weeks. The cells were then fixed with methanol for 20 min at room temperature followed by PBS washing. One percent crystal violet solution was utilized to stain the fixed cells. Photomicrographs of each well were obtained using a camera.

### 4.13. RNA Extraction and qRT-PCR Assay

The TRIzol regent (Invitrogen, Carlsbad, CA, USA) was used to extract the total RNA of tissues and cells according to the manufacturer’s protocol. Then primeScript RT Reagent Kit (Takara, Tokyo, Japan) was used to synthesize cDNA. The expression of EXOSC5 and STAT3 mRNA were determined by SYBR Green PCR Master Mix (Takara, Tokyo, Japan) on real-time PCR system (Roche, Indianapolis, IN, USA) at 95 °C for 30 s, followed by 40 cycles of 95 °C for 5 s and 60 °C for 34 s. ΔCT values were normalized by ACTB levels, and the relative expressions of mRNA to the control were calculated by 2^−ΔΔCT^. The sequence of related genes primers as follow: EXOSC5-F, 5′-ACTTTGCCTGCGAACAGAACC-3′, EXOSC5-R, 5′-CTCTTTGCTGACCTTCACCTC-3′; STAT3-F, 5′-GGCATTCGGAAAGTATTG-3′, STAT3-R, 5′-TCACCCACATTCACTCATT3′; ACTB-F, 5′-CACCAGGGCGTGATGGT-3′, ACTB-R, 5′-CTCAAACATGATCTGGGTCAT-3′.

### 4.14. Western Blot

Cells were seeded into 60 mm cell culture plates at a density of 4 × 10^5^ for about 12~16 h. The cells were lysed using RIPA lysis buffer (Beyotime Institute of Biotechnology, Shanghai, China) containing protease inhibitor cocktail and phenylmethylsulfonyl fluoride (PMSF). The lyses was centrifuged at 1.5 × 10^4^ cells/well for 15 min at 4 °C and the concentration of these supernatants were assayed by BCA Protein assay kit (Beyotime Institute of Biotechnology, Shanghai, China). To separate these proteins, 12% sodium dodecyl sulfate-polyacrylamide gel electrophoresis (SDS-PAGE) was used, and they were then transferred into nitrocellulose (NC) membranes (PALL, Stevenage, UK). Then, the NC membranes were incubated with specific primary antibodies, including Ki67 (ABclonal Technology, Wuhan, China), STAT3 (ABclonal Technology, Wuhan, China), p-STAT3(Cell Signaling Technology, MA, USA), EXOSC5, GAPDH (Cell Signaling Technology, MA, USA) at 1:1000 overnight at 4 °C. In addition, horseradish peroxidase-conjugated immunoglobulin G (IgG) (Cell Signaling Technology, MA, USA) was used to incubate the membranes at 1:5000 for 1 h at room temperature. Immunoreactive protein bands were detected with an Enhanced Chemiluminescence (ECL) Detection Kit (Amersham, Piscataway, NJ, USA). Image J software was used to analyze the relative levels of proteins normalized to the expression of GAPDH.

### 4.15. Statistical Analysis

The data from experiments were presented as the mean ± SD from 3 independent experiments except for the MTT assay. Statistical comparisons between two groups were conducted using Student’s *t*-test, whereas an ANOVA was used to test the significance of differences between multiple groups. *p* < 0.05 was considered significant.

## Figures and Tables

**Figure 1 ijms-23-12161-f001:**
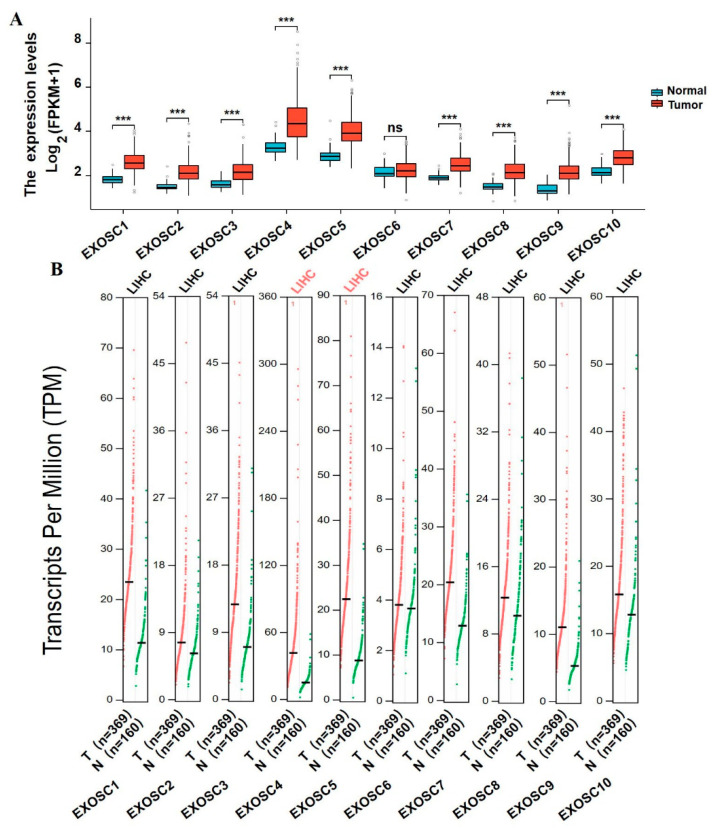
The mRNA expression of EXOSCs in HCC tumor and normal tissues. (**A**) The transcriptional levels of all members in EXOSCs (EXOSC 1–10) from TCGA database. (**B**) The mRNA expression of EXOSCs from GEPIA dataset. *** *p* < 0.001.

**Figure 2 ijms-23-12161-f002:**
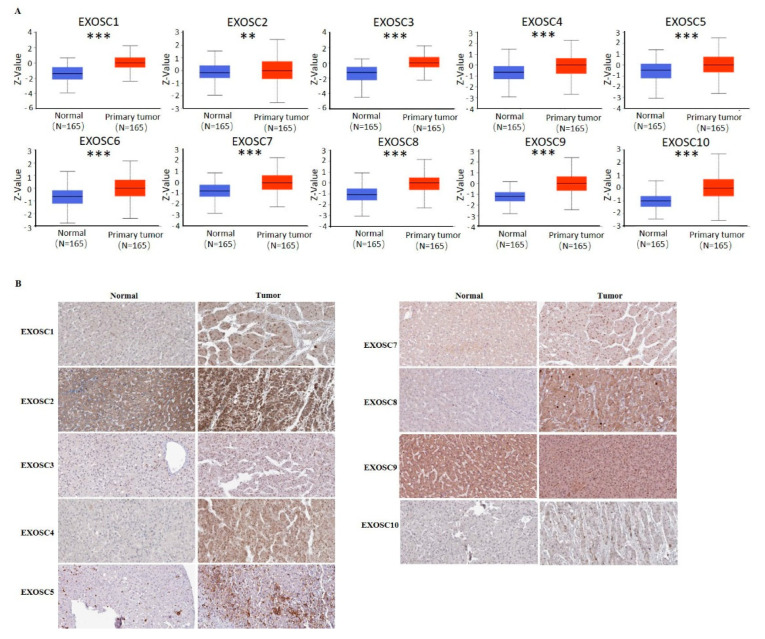
The protein levels of EXOSCs in HCC tumor and normal tissues. EXOSCs in HCC tissues were compared with that in normal tissues by (**A**) Ualcan and (**B**) Human Protein Atlas database. ** *p* < 0.01, *** *p* < 0.001.

**Figure 3 ijms-23-12161-f003:**
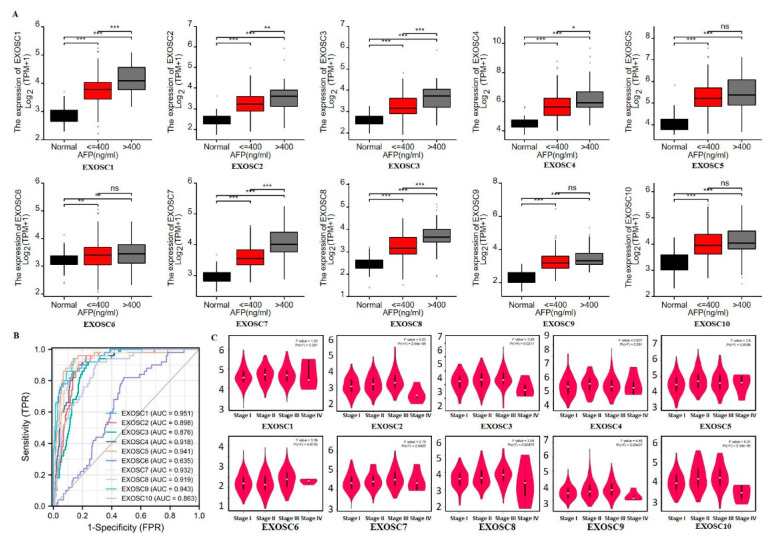
The diagnostic value of EXOSCs in HCC. (**A**) Correlation between upregulated EXOSCs and AFP. (**B**) ROC curve analysis the diagnosis values of EXOSCs in HCC. (**C**) Correlations between EXOSCs expression and tumor stage in HCC patients. * *p* < 0.05, ** *p* < 0.01, *** *p* < 0.001. True positive rate (TPR), False positive rate (FPR).

**Figure 4 ijms-23-12161-f004:**
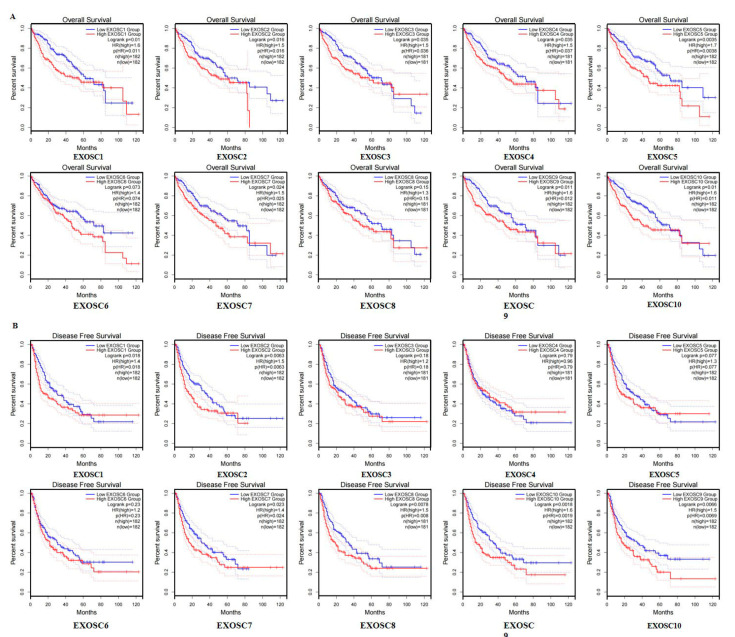
Prognostic potential of the mRNA expression of EXOSCs in HCC (GEPIA). HCC patients with differentially expressed members of EXOSCs were significantly associated with poor (**A**) OS and (**B**) DFS.

**Figure 5 ijms-23-12161-f005:**
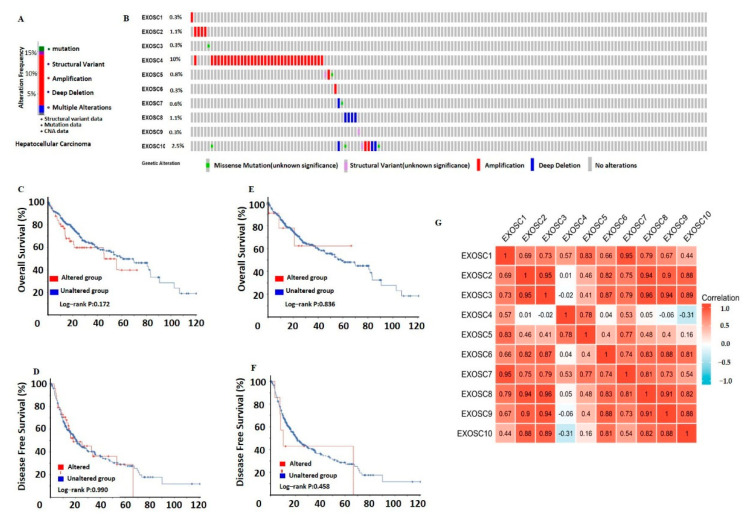
EXOSCs gene mutation and correlation analyses in HCC. (**A**,**B**) summary of alterations in different expressed EXOSCs in HCC. The genetic mutation of EXOSCs. (**C**–**F**) Correlation between the genetic alterations in EXOSC4/10 with OS and DFS of HCC patients. (**G**) The correlation between any two EXOSCs family members.

**Figure 6 ijms-23-12161-f006:**
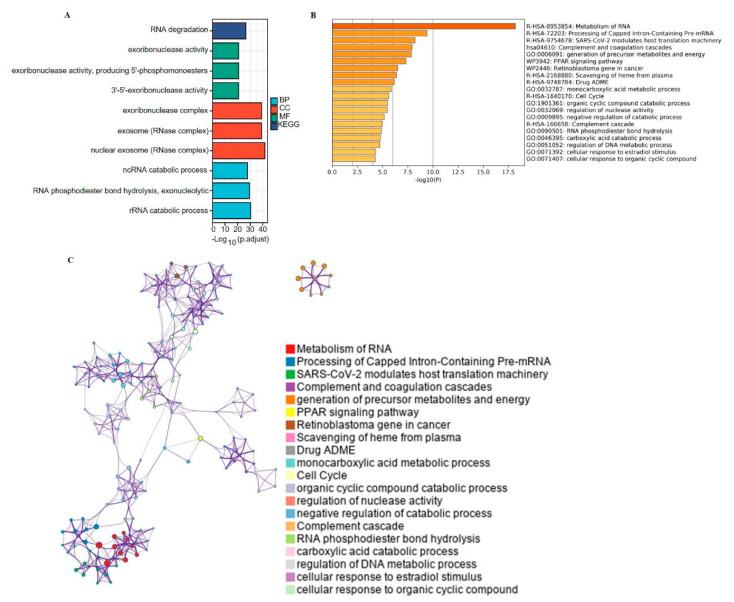
Functional and pathway enrichment analyses of EXOSCs. (**A**) Top significantly enriched GO terms of EXOSCs, including BP, CC, MF and KEGG. (**B**) KEGG enriched terms colored by *p* values. (**C**) Network of enriched terms colored by *p* values.

**Figure 7 ijms-23-12161-f007:**
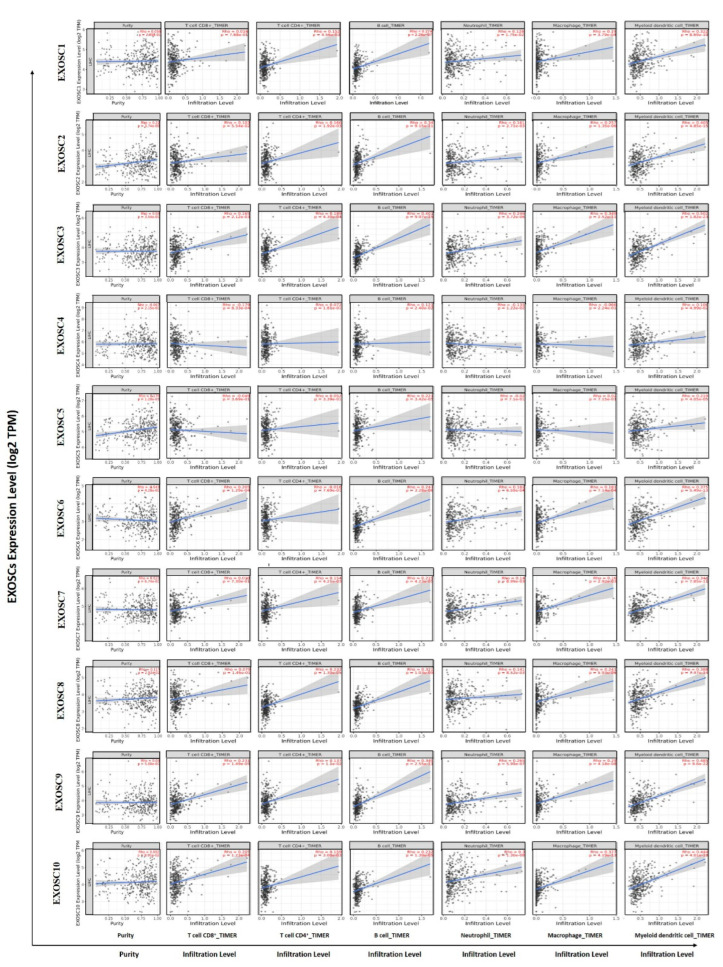
Correlations between differentially expressed EXOSCs and immune cell infiltration. TIMER was used to analyze the correlations between the abundance of immune cells and the expression of EXOSCs family members.

**Figure 8 ijms-23-12161-f008:**
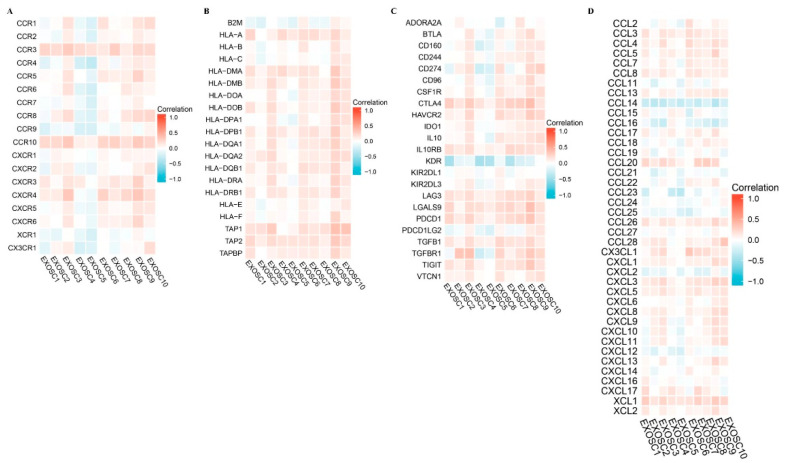
Relationships between the expression levels of EXOSCs and tumor-infiltrating immune cells, immune molecules. The correlation between EXOSCs family expression and, (**A**) immunosuppressive molecules, (**B**) MHC molecule, (**C**) chemokines and (**D**) chemokine receptors in HCC.

**Figure 9 ijms-23-12161-f009:**
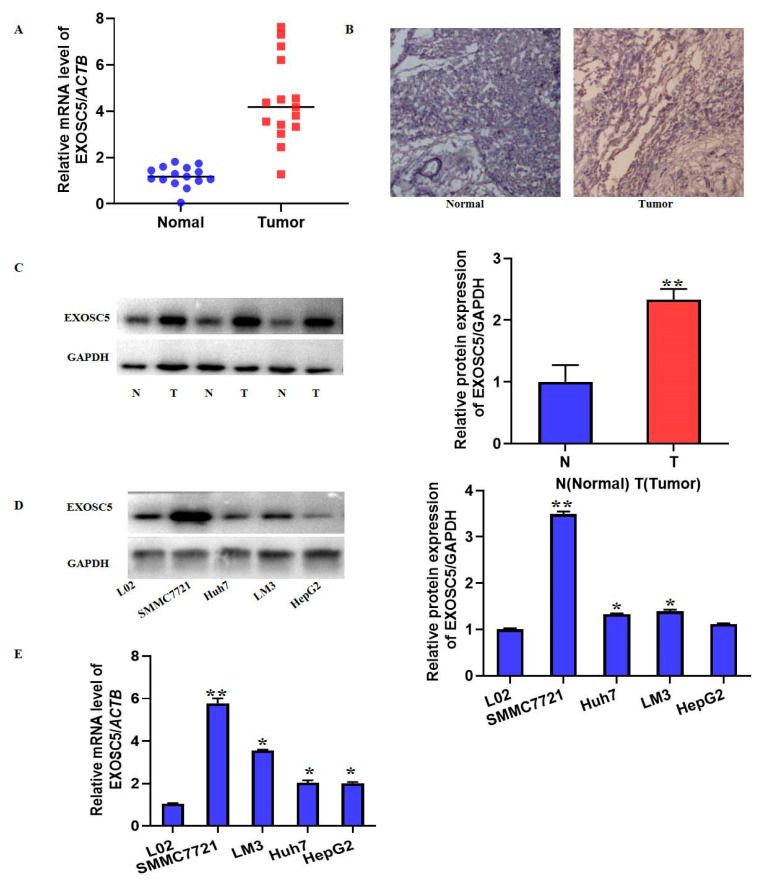
The expression of EXOSC5 in HCC tissue and cell lines. (**A**) The mRNA level of EXOSC5 was detected by qRT-PCR. (**B**) IHC and (**C**) western blot were used to assay the expression of EXOSC5 protein in HCC and adjacent tissues. The protein and mRNA levels of EXOSC5 in L02 hepatocytes and HCC cell lines were detected by (**D**) western blot and (**E**) qRT-PCR. Image J software was used to analyze the relative levels of proteins normalized to the expression of GAPDH. * *p* < 0.05, ** *p* < 0.01, compared with its relative control.

**Figure 10 ijms-23-12161-f010:**
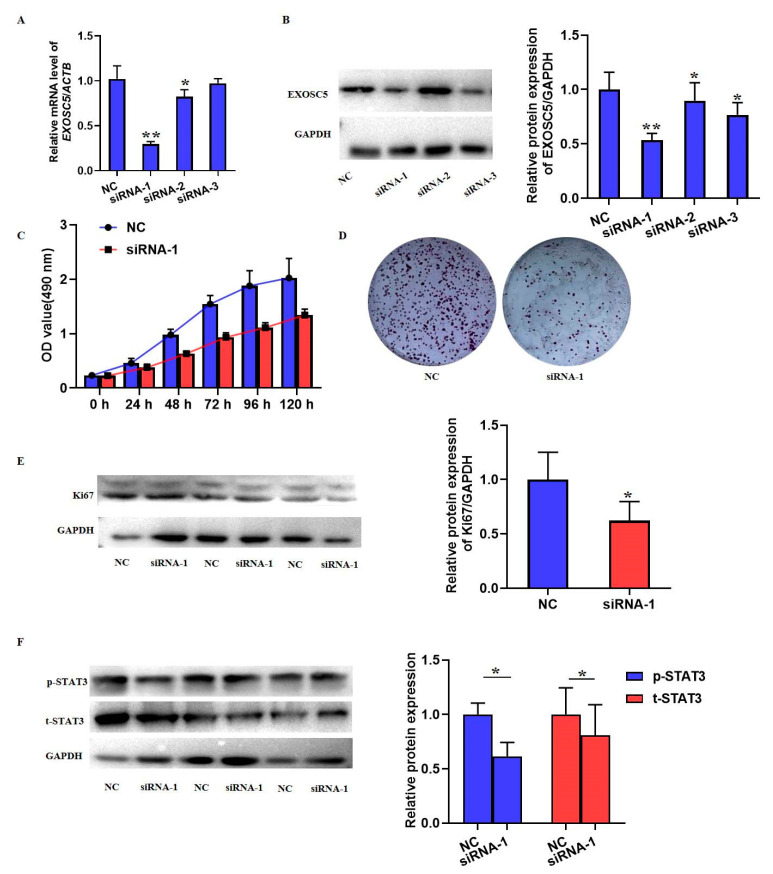
EXOSC5 was related to the cell growth in HCC cells. Three pairs of EXOSC5 siRNAs (siRNA-1, -2 and -3) with different binding sites were employed to silence EXOSC5 in SMMC7721 cells. (**A**) qPCR was performed to detect the mRNA levels of EXOSC5. (**B**) Western blotting was performed to detect the protein levels of EXOSC5. (**C**) MTT assay and (**D**) colony formation assay were used to monitor the cell growth of SMMC7721 cells transfected with EXOSC5 siRNA-1 and the control for the consecutive culture of 120 h. (**E**) The protein expression of Ki67 was detected by western blot. (**F**) The total and phosphorylated STAT3 proteins levels were detected by western blot. Image J software was used to analyze the relative levels of proteins normalized to the expression of GAPDH. * *p* < 0.05, ** *p* < 0.01, compared with its relative control.

**Figure 11 ijms-23-12161-f011:**
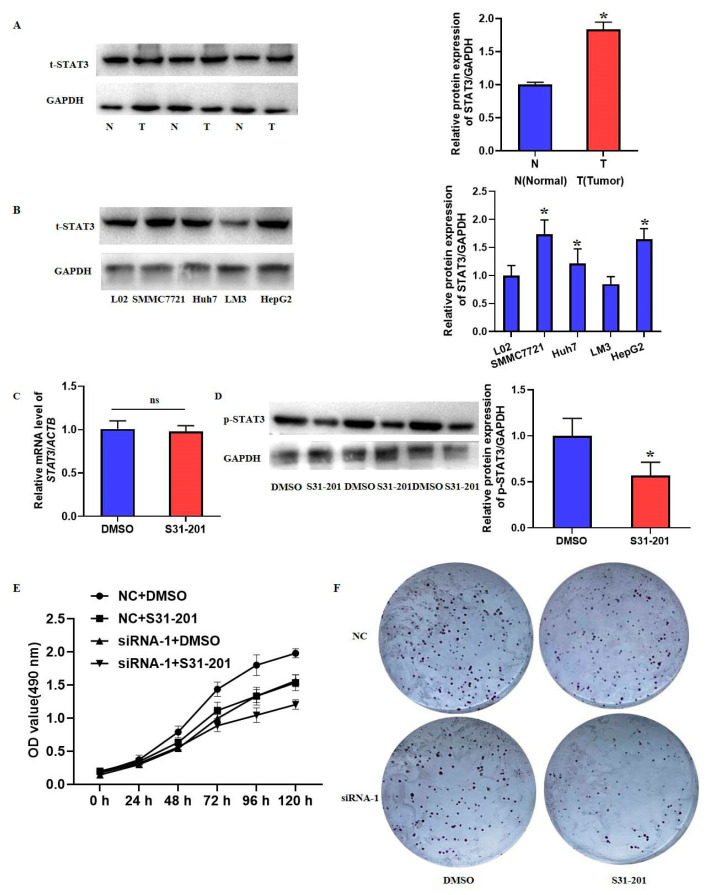
Role of phosphorylated STAT3 in the proliferation of HCC cells. (**A**) The protein expression of STAT3 in HCC and normal tissues. (**B**) The protein expression of STAT3 in L02 hepatocytes and HCC cell lines. (**C**) S31-201 (inhibitor of p-STAT3) was used to treat SMMC7721 cells. The mRNA level of STAT3 was detected by qRT-PCR. (**D**) Western blot was performed to assay the phosphorylation of STAT3. The proliferation of SMMC7721 cell lines were examined by (**E**) MTT assay and (**F**) colony formation assay. Image J software was used to analyze the relative levels of proteins normalized to the expression of GAPDH. * *p* < 0.05, compared with its relative control.

## Data Availability

Publicly available datasets were analyzed in the study, and all data of the experiments has been presented in the manuscript. In addition, the original bands of the proteins have been submitted.

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
