# Peer review of "Integrated Bioinformatic Investigation of EXOSCs in Hepatocellular Carcinoma Followed by the Preliminary Validation of EXOSC5 in Cell Proliferation"

_ijms, 2022, doi:10.3390/ijms232012161_

Round 1

Reviewer 1 Report

A document for review comments was attached.

Author Response

Dear professor,

We sincerely than you for the thoroughly examining our manuscript and providing very helpful comments to guide our revision. We have tried our best to revise the manuscript according to your kind and construction comments and suggestions.

We sincerely hope that this revised manuscript has addressed all your comments and suggestions. We appreciated for your warm work correction will meet with approval.

Yours,

Yujing Zhang,

E-mail: [email protected], [email protected]

Corresponding author: Xin Huang,

E-mail:[email protected]

In the attachment, please find the comments in black, followed by our responses in red. Exact changes in the manuscript are also presented in red font.

Reviewer 2 Report

The reviewer carefully read the manuscript, please consider consulting an English language editing service and find additional comments below:

Line 81: Why does EXOSC5 as intracellular component provide a potential target for HCC treatment, while it is neither specific to HCC cells nor accessible for the immune system?

Fig 1: which statistical test was used, which p-value is indicated by the asterisks?

Fig 1B: resolution is too low, labels not readable

Supplementary figure 1: The provided resolution renders this figure absolutely useless.

Line 111: Which clinical characteristics were considered? Why was EXOSC correlation with AFP analysed? How does AFP relate to the scope of the study?

Fig 2: which statistical test was used, which p-value is indicated by the asterisks?

Fig 3: the resolution is too low, unreadable.

Supplementary figure 2: resolution too low.

Fig 4: the resolution is too low, unreadable.

Line 148: 16.92% of what?

Line 152: Why as expression of EXOSC family members correlated to each other? Which information can be derived from this analysis?

Line 157: What is the conclusion of Figure 5D? Why was an interaction network generated?

Fig 5: the resolution is too low, unreadable.

Section 2.5: Why was GO analysis performed? It is quite useless, as the authors already know in  which processes EXOSC are involved in. This section together with figure 6 can be removed from the manuscript as it is non-informative.

Fig 6: the resolution is too low, unreadable.

Fig 7: useless, as nothing can be read. Increase resolution drastically!

Fig 8: the resolution is too low, unreadable.

Fig 9: the resolution is too low, unreadable.

Fig 10: the resolution is too low, unreadable.

Lin 218: Why were only SMMC7721 cells investigated?

Discussion

Line 286: Why are EXOSC proteins targets for immunotherapy? The immune system cannot access intracellular antigens.

Line 278: One mechanism was investigated in the present study, not several mechanisms.

Line 316-321: This is not a discussion, just general repetition of results which provide no conclusion. The fact that EXOSCs are involved in RNA metabolism etc. is not a finding of this study.

Line 333-335: The statement „Thus, the study provides new insights into understanding the role of EXOSCs in immune cell infiltration of HCC, and provides detailed information to design the new immunotherapies.“ is not supported by the results of this study. The results give no information on roles of EXOSCs (only a correlation – and a correlation does not necessarily mean causation); and the results give absolutely any information on how to design new immunotherapies! A very bold and unscientific claim!

Line 336: Why do you deduce that EXOSC5 impacts HCC progression? Which result(s) lead to this conclusion?

Line 352: This study does not contain in vivo data on the role of EXOSC5 in HCC.

Line 364: Where is the oncogenic role documented? There is an association between EXOSC expression and cancer shown, however, this does not give an idea of onset or progression of a cancer phenotype. Correlation does not imply causation.

Line 365: The statement „but further research is needed to be performed“ is very general and without context an unnecessary space filler. Which aspects need to be researched?

Methods

4.1 Please describe the procedures in more detail, in particular the data selection and processing. By now, the paragraph only describes where the data come from. Based on this alone, the results can not be replicated by other researchers.

Line 383: Why exactly 400 ng/ml?

Line 452: Please provide the sequence of EXOSC5 siRNA!

Line 458: S21-301 or S21-201?

Line 486: What is the unit of 1.5x10^4?

Statistical analyses: Were the data tested for normality? Otherwise non-parametric tests are required.

General questions

Why and how do higher EXOSC levels promote cancer?

What is the novelty of the present study, as the authors referenced some works on elevated EXOSC5 in a plethora of tumors in the discussion?

How many patients or replicates were involved in the experiments or analyses. This is nowhere indicated.

The provided original images of the Western Blots are almost identical to the cropped images shown in the figures, therefore, the „original images“ are not really original. They could have been cut out and been „re-used“ from anywhere. None of the original images show molecular weight markers. Highly unreliable.

The introduction should more clearly outline the hypotheses and aims of the study and highlight, why this study is novel as the authors reference a couple of studies reporting elevated EXOSC expression in cancer. It seems that is study is just replicating existing studies.

The dicussion does not include any statements of weaknesses and/or limitations of the study. The repeatedly made claim that the results can be used to develop (immune) therapies against HCC is unsubstantiated and not supported by the data.

Author Response

Dear professor,

We sincerely than you for the thoroughly examining our manuscript and providing very helpful comments to guide our revision. We have tried our best to revise the manuscript according to your kind and construction comments and suggestions.

We sincerely hope that this revised manuscript has addressed all your comments and suggestions. We appreciated for your warm work correction will meet with approval.

Yours,

Yujing Zhang,

E-mail: [email protected] , [email protected] 

Corresponding author: Xin Huang,

In the attachment, please find the comments in black, followed by our responses in red. Exact changes in the manuscript are also presented in red font.

Round 2

Reviewer 1 Report

All comments were addressed well.

Author Response

Dear professor,

We sincerely thank you for the thoroughly examining our revised manuscript. we are also happy that the correction could meet with approval.

Once again, thank you very much.

Yours,

Yujing Zhang,

E-mail: [email protected], [email protected] 

Corresponding author: Xin Huang,

Reviewer 2 Report

The reviewer thanks the authors for addressing the comments, the manuscript improved a lot in quality! However, some issues are still open which can be managed in a minor revision:

The figures are provided as larger images, however with the same resolution and the textlabels are still largely not readable. Please increase the RESOLUTION and/or the font size of textlabels and don’t simply scale up the image! This applies to figures 1, 2A, 3C, 4, 5, and 7.

Fig 5H protein interaction network: It is known that the displayed proteins are involved in a multimeric protein complex interacting with each other, therefore it is quite unnecessary to draw a protein interaction network as no further information can be derived from it, at least none is reported in the text. In particular, when the network is overcrowded with connector lines and heavily underresolved pixelwise. The other panels of Fig 5 are informative, so C-F should be arranged as square and G placed to the right.

Concerning the change from “in vivo” to “in situ” in the discussion, “in situ” is still not true, the data from cell culture based experiments are “in vitro” results. “in situ” would mean the experiments were performed on cells within a tissue context. If the results from bioinformatic analyses are meant, “in silico” would be the appropriate term.

Thank you for explaining why 400 ng/ml AFP was used – please add the informative explanation from <<Guidelines for Diagnosis and Treatment of Primary Liver Cancer (2022)>> into the manuscript text, as it is interesting for the readers.

Concerning the revised section “Our study documented the role and underlying mechanism of EXOSC5 in promoting HCC proliferation and tumorigenesis, but further research about the function and molecular mechanisms of EXOSC5 in HCC is required.” – Why is further research required, what is still not known as you claim to have identified the mechanism of EXOSC5-mediated HCC proliferation?

Author Response

Dear professor,

We sincerely thank you for the thoroughly examining our manuscript and providing very helpful comments to guide our revision. We have tried our best to revise the manuscript according to your kind and construction comments and suggestions.

We sincerely hope that this revised manuscript has addressed all your comments and suggestions. We appreciated for your warm work correction will meet with approval.

Yours,

Yujing Zhang,

E-mail: [email protected], [email protected] 

Corresponding author: Xin Huang,

Below, please find the comments in black, followed by our responses in red italics. Exact changes in the manuscript are also presented in red italics.
